



# Two months of disdrometer data in the Paris area

Auguste Gires[1], Ioulia Tchiguirinskaia[1], and Daniel Schertzer[1]

[1]HMCo, Ecole des Ponts, UPE, Champs-sur-Marne, France

*Correspondence to:* Auguste Gires (auguste.gires@enpc.fr)

**Abstract.**

The Hydrology, Meteorology and Complexity laboratory of Ecole des Ponts ParisTech (hmco.enpc.fr) makes available a data set of optical disdrometers measurements coming from a campaign involving three collocated devices from two different manufacturers relying on different underlying technologies (one Campbell Scientific PWS100 and two OTT Parsivel$^2$). The campaign took place on January-February 2016 in the Paris area (France). Disdrometers give access to the size and velocity of drops falling through the sampling area of the devices of roughly few tens of cm$^2$. It enables to estimate the drop size distribution and further study rainfall micro-physics, kinetic energy or radar quantities for example. Raw data, i.e. basically a matrix containing a number of drops according to classes of size and velocity, along with more aggregated one such rain rate or drop size distribution with filtering is available.

Link to the data set: https://zenodo.org/record/1125583

*Copyright statement.* TEXT

## 1 Introduction

Disdrometers enable to access not only rainfall depth as rain gauges, but also the size and velocity of falling drops. Their operational and research use is quickly increasing. Historically, impact disdrometers were the first widely used. They measure the noise generated by the impact of falling drops (Joss and Waldvogel, 1967). Now the most commonly used operationally are optical ones. They are made of a transmitter generating one or several laser sheet(s) and receiver(s) measuring either the occluded light (Loffler-Mang and Joss, 2000; Battaglia et al., 2010; Delahaye et al., 2006; Frasson et al., 2011) or the refracted light (Ellis et al., 2006) from the drops falling through a sampling area of roughly few tens of cm$^2$. The received signal is then processed to estimate the size (more precisely an equivolumic diameter, i.e. the diameter of a spherical drop having the same volume) and fall velocity of the hydrometeor. More sophisticated disdrometers such the 2D Video one can also give access to images of the falling hydrometeors (Kruger and Krajewski, 2002). Numerous studies have been carried out to compare the output of various types of disdrometers usually with the more conventional rain gauges (Miriovsky et al., 2004; Krajewski et al., 2006; Frasson et al., 2011; Thurai et al., 2011).





The Drop Size Distribution (DSD), denoted $N(D)$, can be computed from disdrometer data. It is expressed in $m^{-3}.mm^{-1}$ and $N(D)dD$ is the number of drops per unit volume (in m$^{-3}$) with an equivolimic diameter between $D$ and $D+dD$ (in $mm$). The DSD enables to study rainfall microphysics through for example the computation of quantities such as the total drop concentration or the mass weighed diameter characterizing the overall DSD (Pruppacher and Klett, 1997; Jaffrain and

5 Berne, 2011). From the DSD it is also possible to estimate equivalent local pointwise radar quantities and hence study and possibly help improve radar rainfall retrieval algorithms. This approach has been implemented by numerous authors (Jaffrain and Berne, 2012a; Leinonen et al., 2012; Ryzhkov et al. , 2005; Verrier et al., 2013; Gires et al., 2015). It should be mentioned that it relies on strong unrealistic hypothesis, notably the homogeneity of the DSD within a radar bin (see Gires et al., 2017b for a discussion of the limitations of this approach). The DSD also enables to compute the kinetic energy of rainfall which

10 is critical to the understanding and modelling of soil erosion (see van Dijk et al., 2002, for a review). Studies of the relation between rainfall intensity and kinetic energy using disdrometers have been carried out with actual rainfall (Angulo-Martínez and Barros, 2015; Angulo-Martínez et al., 2016) or artificial one (Meshesha et al., 2016). DSD also has an impact on the spread of crop disease (Huber and Gillespie, 1992) notably through the dispersal of pathogens through splashing (Walklate, 1989; Walklate et al., 1989).

15 Given the numerous potential applications of DSD data, the Hydrology, Meteorology and Complexity laboratory of Ecole des Ponts ParisTech (HMCo-ENPC) believes it is relevant to make available the data from a two month measurement campaign involving three collocated optical disdrometers from two manufacturers. Devices, data processing and campaign period is presented in section 2. The corresponding data base and available tools are presented in section 3.

## 2 Data and methods

20 **2.1 Brief description of the devices' functioning**

The two devices are optical disdrometers that do not work on the same principle. The goal of this section is only to briefly explain how the two devices work and highlight the main differences. The interested reader is then referred to papers and manufacturer documentation for more information.

The OTT Parsivel[2] is made of a transmitter generating a laser sheet and a receiver aligned with the transmitter. When a drop

25 falls through the sampling area of roughly 50 cm$^2$, the laser beam is partially occluded and the intensity of the received signal decreases. Then the size (more precisely the equivolumic diameter) and fall velocity of the drop is assessed from the amplitude and duration of the decrease in received intensity. An ellipsoidal shape model for the drops with a standard relation between axis ratio and equivolumic diameter is assumed in the process. More details can be found in Battaglia et al. (2010) or in the device documentation (OTT, 2014).

30 The PWS100 configuration is not the same. It is made of a transmitter that generates 4 horizontal parallel light sheets, and two receptors which are not aligned with the transmitter. One is set on a vertical plane with an angle $\theta_D$ (= 20$^o$) and the other is set on a horizontal plane with the same angle $\theta_D$. When a drop falls through the sampling area of size S = 40 cm$^2$ a portion of the light is refracted and reaches the receptors. The signal received by each receptor contains four consecutive peaks associated





with each laser sheet. From the delay between these peaks, the fall velocity of the hydrometeor is assessed. Then due to the refraction inside the drop, the signal will reach the vertical receiver slightly before the horizontal one. From this time shift the diameter of the drop can be estimated. Computations are carried out assuming a spherical shape and a correction accounting for oblateness is then implemented. More details can be found in Ellis et al. (2006) or in the device documentation (Campbell-

Scientific-Ltd, 2012). Actually authors found possible to improve the oblateness correction of the PWS100 rationale (Gires et al., 2017a) and used it in previous studies (Gires et al., 2015).

## 2.2   Available output and data processing

The main output provided by both disdrometers is a matrix containing the number of drops recorded during the time step $\Delta t$ according to classes of equivolumic diameter (index $i$ and defined by a centre $D_i$ and a width $\Delta D_i$ expressed in $mm$) and fall

velocity (index $j$ and defined by a centre $v_j$ and a width $\Delta v_j$ expressed in $m.s^{-1}$). The measurement time step $\Delta t$ is equal to 30 $s$ for this data set. The classes for each devices are shown in Table 1 and  2. In practice, the solution suggested by authors to improve oblateness correction (Gires et al., 2017a), simply consists in changing the center and width defining the diameter classes for the PWS100. A new suggested table for rain drop was hence added to Table 1. Values and figures presented in this paper are obtained with the new correction. However it does not affect the retrieved matrixes meaning that it is up to the user to

decide whether to use it or not. It should be mentioned that for the PWS100, data is measured only 9/10th of the time, hence to have comparable values, one should multiply by 10/9 the number of drops in each class for this device. All outputs presented in this paper and in the database take into account this correction. However it is not implemented on the raw data (basically the matrices) maide available.

From this matrix, it is then possible to compute the rain rate (in mm.h$^{-1}$) for each time step as :

$$R = \frac{\pi}{6\Delta t} \sum_{i,j} \frac{n_{i,j} D_i^3}{S_{eff}(D_i)} \tag{1}$$

where $S_{eff}(D_i)$ is the sampling area of the device. In the data presented in the paper, it is slightly modified according to the drop size to account for edge effects for large drops. For the Parsivel[2], we used $S_{eff}(D_i) = L(W - \frac{D_i}{2})$ where $L = 180$ mm and $W = 30$ mm are respectively the length and width of the sampling area ($LW = 54$ cm$^2$) (OTT, 2014). Due to its different configuration, the PWS100 is not affected by this issue and a constant $S_{eff}$ equal to 40 cm$^2$ is considered (Campbell-

Scientific-Ltd, 2012). Again the user has access the raw data (i.e. the matrix), so he/she can decide on whether to use this correction.

A discrete DSD, $N(D_i)$, for a given time step can also directly be computed from the raw matrix as :

$$N(D_i) = \frac{1}{S_{eff}(D_i)\Delta D_i \Delta t} \sum_j \frac{n_{i,j}}{v_j} \tag{2}$$

$N(D_i)\Delta D_i$ gives the number of drops with a diameter in the class $i$ per unit volume (in m$^{-3}$).

For the data presented in this paper, we also included for the values presented in this paper a filter suggested by various author (Kruger and Krajewski, 2002; Thurai and Bringi, 2005; Jaffrain and Berne, 2012b) to remove drops that, according to

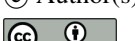



their size and velocity, are considered as non-meteorological measurements. Practically hydrometeors whose velocity differs of more than 60% from the terminal fall velocity expected from its diameter and Beard's formula (Beard, 1977) are removed. Again, this filter is not implemented in the available raw matrixes, so it user are obviously free to use it or not.

The temperature in $^oC$ is also provided by both devices. For the Parsivels[2], the resolution of the sensor is only of $1^oC$ while for the PWS100 it is a dedicated sensor in the meteorological shelter visible on the mast of the PWS100 on Fig. 1.

### 2.3 Measurement period

The three disdrometers whose data is available in this paper were located on the roof of the Carnot building of the Ecole des Ponts ParisTech campus, in the the Paris area (Fig. 1). They are part of the TARANIS observatory (exTreme and multi-scAle RAiNdrop parIS observatory, Gires et al., 2015) of the Fresnel Platform of Ecole des Ponts ParisTech (https://hmco.enpc.fr/Page/Fresnel-Platform/en). As it can be seen on Fig. 1, the two Parsivels[2] (black devices) are oriented perpendicularly. For this campaign denoted Carnot_1, the data was collected during the months of January and February 2016 during which there was very little missing data. More precisely, over these 60 days, there are 2398, 2399 and 3180 missing times steps (roughly 20 to 26 hours) for respectively the Parsivel[2] #1, the Parsivel[2] #2 and the PWS100. For the Parsivels[2], it corresponds to dry periods during which the data was uploaded from the computer collecting the data simultaneously for the three devices. The additional ones for the PWS100 (which are spread over the period) corresponds to time steps during which the retrieval was not possible for unknown reasons. It should be mentioned that during these additional time steps both Parsivels[2] collected less than .1 mm of rain.

Figure 2 displays the temporal evolution of the rain rate and cumulative rainfall depth during the two month period. It can be seen that despite the correction implemented the total rainfall depth for PWS100 (104 mm) remains significantly superior to the one measured with Parsivel[2] #1 (86 mm) and Parsivel[2] #2 (84 mm). Such total depth are common for the period and the area. Indeed according to Meteo France, the French official meteorological agency, the climatological average over these two months is of 92.2 mm in Paris (18.25 Km West from disdrometers location) and 102.7 mm in Melun (32 Km South from disdrometers location) (source http://www.meteofrance.com/climat).

### 3 Data base

This section contains a description of the data base content along with some available scripts. The data base is organized as follow:

disdrometers_data_base/

    Raw_data_zip/

        Pars1/

        Pasr2/

        PWS/

    Each folder contains the files for its disdrometers.



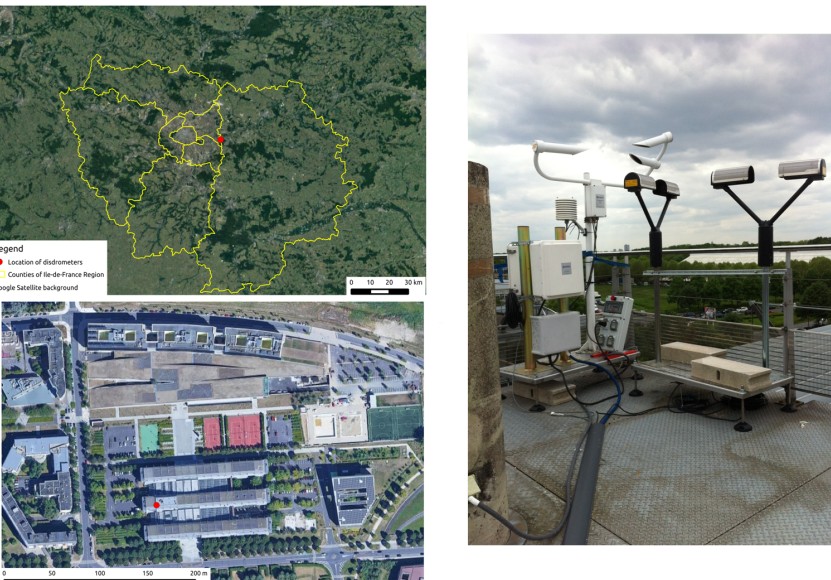

**Figure 1.** Location of the disdrometers in the Paris area (top left), on the roof of the Carnot building (right) at the ENPC campus (bottom left)

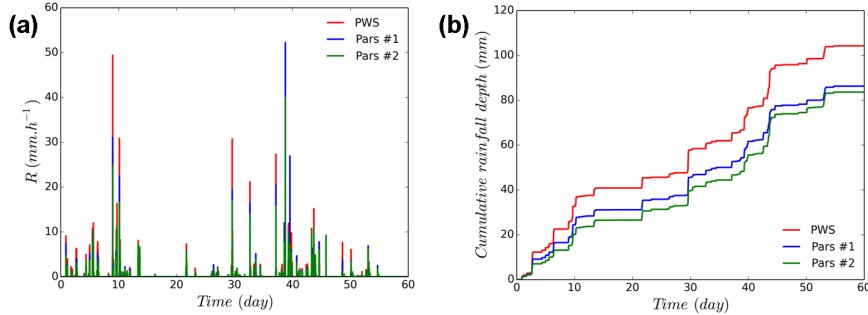

**Figure 2.** Temporal evolution of the rain rate and cumulative rainfall depth over the months of the measurement campaign for the three disdrometers

The name is Raw_DisdroName_YYYYMMDD.zip (ex : Raw_pars1_20160110.zip)

Daily_data_python/

Pars1/

Pasr2/

5      PWS/

Each folder contains the files for its disdrometers.

The name is DisdroName_raw_data_YYYYMMDD.csv (ex : Pars1_raw_data_20160110.npy)



Daily_data_csv/

Pars1/

Pasr2/

PWS/

Each folder contains the files for its disdrometers.

The name is DisdroName_daily_data_YYYYMMDD.csv (ex : Pars1_daily_data_20160110.npy)

Calendars/

Data_5_min/ (one file per day ex: R_5_min_Carnot_1_2016_01_10_00_00_00__2016_01_10_23_59_30.csv)

Data_30_sec/ (one file per day ex: R_30_sec_Carnot_1_2016_01_10_00_00_00__2016_01_10_23_59_30.csv)

Quicklooks/ (one file per day ex : Quicklook_Carnot_1_2016_01_10_00_00_00__2016_01_10_23_59_30.png

Calendar_data_5_min_Carnot_1.html

Calendar_data_30_sec_Carnot_1.html

Calendar_Carnot_1.html

Python_scripts/

It contains the python scripts (and associated files) to generate and use this data base.

Read_me.txt

It contains a short description of the Taranis data base.

## 3.1   Calendars

This folder contains a .html file ('Calendar_Carnot_1.html') providing an overview of the measurement campaign. Fig. 3
displays a snapshot of it. It enables to quickly identify the most interesting days according the aim of the user. By click-
ing on any day, one can access a quicklook of the corresponding day. Fig. 4 shows an example. This quicklook provides an
overview of the day according to the measurements of the three disdrometers with (1) The temporal evolution of the rain
rate (upper left); (2) The temporal evolution of the cumulative rainfall depth (upper right); (3); The temporal evolution of
the DSD $N(D)$ (middle left); (4) Indication of the missing data if any (middle right; in the example there are two missing
time steps for the PWS100 and none for the Parsivels[2]) (5) A map of the number of drops according the velocity and size
classes (middle right; the solid black line is the curve corresponding to the relation between the terminal fall velocity of drops
as a function of their equivolumic diameter obtained by Lhermitte et al., 1988) (middle right); (6) $N(D)D^3$ as a function
of $D$ (lower left, it was chosen to plot $N(D)D^3$ and not simply $N(D)$ because it is proportional to the volume of rain ob-
tained according to the drop diameter hence providing the reader a greater immediate insight of the influence of the various
drops size on the observed rainfall event); (7) The temporal evolution of the temperature (lower right). The quicklooks are
stored in the folder Quicklooks/ and can be accessed directly there. Their names are Quicklook_Carnot_1_ followed by the
date of start and end of the corresponding day in string format. For example the one displayed in Fig. 4 is called 'Quick-
look_Carnot_1_2016_01_10_00_00_00__2016_01_10_23_59_30.png'. Local time are used.





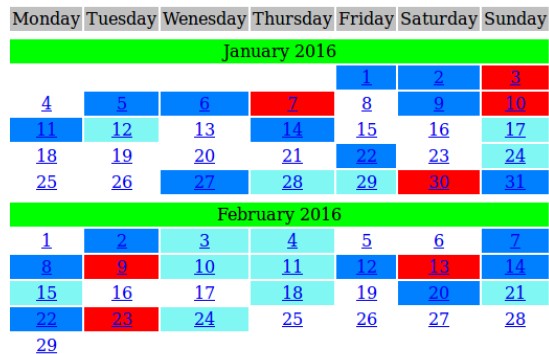

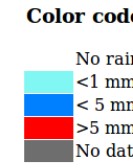

**Figure 3.** Snapshot of the calendar summarizing the January-February 2016 campaign on the roof of the Carnot building

The file 'Calendar_R_30_sec_Carnot_1.html' is a calendar similar to the previous one except that the links give access to the rain rates (in $mm.h^{-1}$) for each 30 stime step of the day in .csv format. The files are located in the folder Data_30_second/ and named in a similar way as the quicklooks (example : 'R_30_sec_Carnot_1_2016_01_10_00_00_00__2016_01_10_23_59_30.csv'). The format is : (i) One line per 30 $s$ time step starting on YYYY-MM-DD 00:00:00 (local time); (ii) In each line, values for the three disdrometers are separated with semi column and the order is PWS;Pars#1;Pars#2; (iii) Missing data are noted as "nan".

The file 'Calendar_R_5_min_Carnot_1.html' is a calendar similar to the previous one except that the links give access to the rain rates (in $mm.h^{-1}$) for each 5 min time step of the day in .csv format. The files are located in the folder Data_5_min are named in a similar way as the quicklooks (example : 'R_30_sec_Carnot_1_2016_01_10_00_00_00__2016_01_10_23_59_30.csv'). The format is the same as for the 30 s time step data.

## 3.2 Daily_data_csv/

This folder contains a folder for each of the three disdrometers. Each of these folders contains daily file with the most relevant data measured by the device, i.e. the full matrix of drops according to classes of size and velocity and the temperature. A file is typically called 'Pars1_daily_data_2016_01_10_00_00_00__2016_01_10_23_59_30.csv' meaning the disdrometer name and start and end of the period corresponding to the data is easily visible for the user. The format is the following : (i) One line per time step; (ii) For each line : Date (YYYY-MM-DD HH:MM:SS); number of drops per class of velocity and size (1st size class





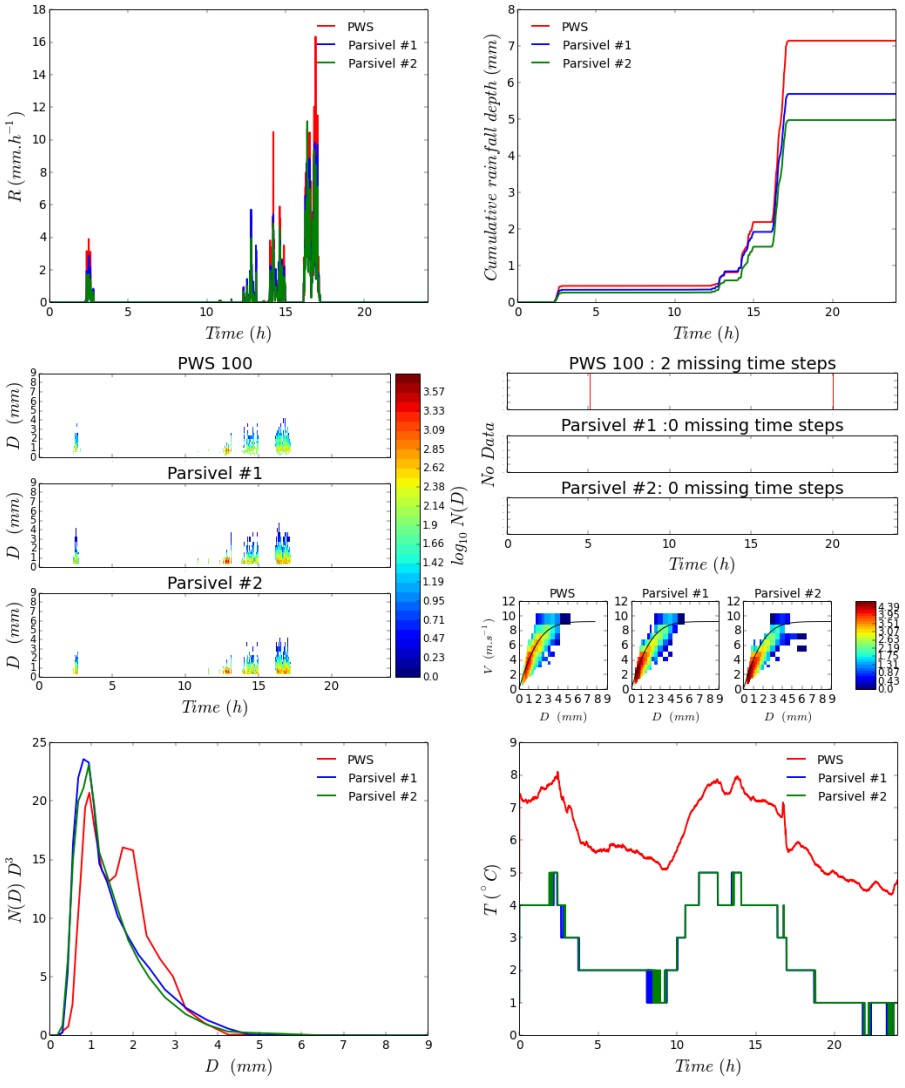

**Figure 4.** Quicklook of the disdrometer data available on 10 January 2016

- 1st velocity class,1st size class - 2nd velocity class, 1st size class - 2nd velocity class, ... , 2nd size class - 1st velocity class...) separated by comas (34 * 34 classes for PWS data and 32 * 32 classes for Parsivel data); Temperature (in $^oC$). ex : 2016-01-10 17:27:00;0.0,0.0,0.0......,0.0;7 (iii) missing data are indicated as nan. It should be reminded here that the number of drops for each class should be multiplied by 10/9 for the PWS100 because it only measures 9/10th of the time. These files are text file

5  that can be read by any software.

### 3.3 Daily_data_python/

This folder contains a folder for each of the three disdrometers. Each of these folders contains a daily file as a list of the data collected by the devices. It is store in .npy format readable with the help of Python 3. The precise content of the list according to the device can be found in the 'Read_me_v1.txt' file. These files are used by the Python scripts made available.

### 3.4 Raw_data_zip/

This folder contains a folder for each of the three disdrometers. Each of these folders contains a zip file for each of the day. This .zip file contains the data directly collected from the disdrometer. There is one text file for each 30 s time step. The precise format of these fields can be found in the Python scripts in the heading of the corresponding functions. This corresponds to the raw data. It is made available for the expert users, but in practice it is believed that the .csv file or the python scripts should be sufficient for most of the users.

### 3.5 Python_scripts/

This section contains some Python scripts that can be used to carry out some initial analysis and data treatement with the data base. The tools are located in the script 'Tools_data_base_use_v3.py'. The main functions are (only a short description is given here - more details, including precise description of the inputs and outputs of the functions, are provided as comments in each script) :

- Quicklook_and_R_series_generation_Carnot_1 : generating a quicklook image and the corresponding 30 s and 5 min rain rate time series for a given rainfall event for the Carnot_1 campaign.

- extracting_one_event_Pars_Rad : reading daily.npy files and generating three lists (one for each disdrometer) containing all the data that can be analyzed for the Carnot_1 campaign.

- exporting_full_matrix : reading daily.npy files and exporting full matrix in .csv files for a given disdrometer and event

- exporting_R : reading daily.npy files and exporting R in .csv files for a given disdrometer and event

- exporting_T : reading daily.npy files and exporting T in .csv files for a given disdrometer and event

Commented examples of use of the functions can be found in the scripts : 'Example_of_use_data_base_Carnot_1.py'. Note that Python 3 (www.python.org) is required because the .npy files containing the data were saved using Python 3.

## 4 Data access and terms of use

The 30 s disdrometer data from a two month measurement campaign with collocated devices in the Paris area is presented in this paper. Raw data along with python formated data with corresponding scripts are described. The Hydrology, Meteorology and



Complexity laboratory of Ecole des Ponts ParisTech (HMCo-ENPC) has made this data set available at https://zenodo.org/record/1125583.
The following citation should be used for every use of the data :

– For this paper : INCLUDE CITATION of this paper

– For the data base : Gires A., Tchiguirinskaia I., Schertzer D., 2017. Data for"Two months of disdrometer data in the Paris
5    area". https://zenodo.org/record/1125583

This data set is available for download free of charge. License terms apply. These disdrometers and others will be used by
HMCo-ENPC in other campaigns and regular updates of the data base are to be provided through its website (hmco.enpc.fr)

*Competing interests.* The authors declare that they have no conflict of interest.

*Acknowledgements.* The authors greatly acknowledge partial financial support from the Chair of Hydrology for Resilient Cities (sponsored
10  by Veolia) of the Ecole des Ponts ParisTech, EU NEW INTERREG IV RainGain Project (www.raingain.eu), EU Climate KIC Blue Green
Dream (www.bgd.org.uk), and the Ile-de-France region Rad@IdF Project.
    The authors thank Serge Botton (from the Departement Positionnement Terrestre et Spatial of the ENSG) for facilitating access to the roof
where the disdrometers are installed.





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



**Table 1.** Definition of the classes of particule size and velocity for the PWS100

| | Particle diameter classes | | | | | Particle velocity classes | |
|---|---|---|---|---|---|---|---|
| Class | Diameter (mm) | Width (mm) | Corr. diam. (mm) | Corr. width (mm) | Class | Velocity (m.s$^{-1}$) | Width (m.s$^{-1}$) |
| 1 | 0.05 | 0.1 | 0.05 | 0.1 | 1 | 0.05 | 0.1 |
| 2 | 0.15 | 0.1 | 0.15 | 0.1 | 2 | 0.15 | 0.1 |
| 3 | 0.25 | 0.1 | 0.25 | 0.1 | 3 | 0.25 | 0.1 |
| 4 | 0.35 | 0.1 | 0.35 | 0.1 | 4 | 0.35 | 0.1 |
| 5 | 0.45 | 0.1 | 0.45 | 0.1 | 5 | 0.45 | 0.1 |
| 6 | 0.55 | 0.1 | 0.55 | 0.1 | 6 | 0.55 | 0.1 |
| 7 | 0.65 | 0.1 | 0.65 | 0.1 | 7 | 0.65 | 0.1 |
| 8 | 0.75 | 0.1 | 0.75 | 0.1 | 8 | 0.75 | 0.1 |
| 9 | 0.85 | 0.1 | 0.85 | 0.1 | 9 | 0.85 | 0.1 |
| 10 | 0.95 | 0.1 | 0.95 | 0.1 | 10 | 0.95 | 0.1 |
| 11 | 1.1 | 0.2 | 1.07 | 0.16 | 11 | 1.1 | 0.2 |
| 12 | 1.3 | 0.2 | 1.24 | 0.16 | 12 | 1.3 | 0.2 |
| 13 | 1.5 | 0.2 | 1.41 | 0.16 | 13 | 1.5 | 0.2 |
| 14 | 1.7 | 0.2 | 1.58 | 0.16 | 14 | 1.7 | 0.2 |
| 15 | 1.9 | 0.2 | 1.74 | 0.16 | 15 | 1.9 | 0.2 |
| 16 | 2.2 | 0.4 | 1.99 | 0.32 | 16 | 2.2 | 0.4 |
| 17 | 2.6 | 0.4 | 2.31 | 0.31 | 17 | 2.6 | 0.4 |
| 18 | 3.0 | 0.4 | 2.62 | 0.31 | 18 | 3.0 | 0.4 |
| 19 | 3.4 | 0.4 | 2.93 | 0.3 | 19 | 3.4 | 0.4 |
| 20 | 3.8 | 0.4 | 3.24 | 0.3 | 20 | 3.8 | 0.4 |
| 21 | 4.4 | 0.8 | 3.68 | 0.58 | 21 | 4.4 | 0.8 |
| 22 | 5.2 | 0.8 | 4.26 | 0.55 | 22 | 5.2 | 0.8 |
| 23 | 6.0 | 0.8 | 4.8 | 0.52 | 23 | 6.0 | 0.8 |
| 24 | 6.8 | 0.8 | 5.3 | 0.48 | 24 | 6.8 | 0.8 |
| 25 | 7.6 | 0.8 | 5.77 | 0.44 | 25 | 7.6 | 0.8 |
| 26 | 8.8 | 1.6 | 6.4 | 0.76 | 26 | 8.8 | 1.6 |
| 27 | 10.4 | 1.6 | 7.07 | 0.57 | 27 | 10.4 | 1.6 |
| 28 | 12.0 | 1.6 | 7.54 | 0.36 | 28 | 12.0 | 1.6 |
| 29 | 13.6 | 1.6 | 7.78 | 0.12 | 29 | 13.6 | 1.6 |
| 30 | 15.2 | 1.6 | | | 30 | 15.2 | 1.6 |
| 31 | 17.6 | 3.2 | | | 31 | 17.6 | 3.2 |
| 32 | 20.8 | 3.2 | | | 32 | 20.8 | 3.2 |
| 33 | 24.0 | 3.2 | | | 33 | 24.0 | 3.2 |
| 34 | 27.2 | 3.2 | | | 34 | 27.2 | 3.2 |



**Table 2.** Definition of the classes of particule size and velocity for the Parsivel[2]

| | Particle diameter classes | | | Particle velocity classes | |
|---|---|---|---|---|---|
| Class | Diameter (mm) | Width (mm) | Class | Velocity (m.s$^{-1}$) | Width (m.s$^{-1}$) |
| 1 | 0.062 | 0.125 | 1 | 0.05 | 0.1 |
| 2 | 0.187 | 0.125 | 2 | 0.15 | 0.1 |
| 3 | 0.312 | 0.125 | 3 | 0.25 | 0.1 |
| 4 | 0.437 | 0.125 | 4 | 0.35 | 0.1 |
| 5 | 0.562 | 0.125 | 5 | 0.45 | 0.1 |
| 6 | 0.687 | 0.125 | 6 | 0.55 | 0.1 |
| 7 | 0.812 | 0.125 | 7 | 0.65 | 0.1 |
| 8 | 0.937 | 0.125 | 8 | 0.75 | 0.1 |
| 9 | 1.062 | 0.125 | 9 | 0.85 | 0.1 |
| 10 | 1.187 | 0.125 | 10 | 0.95 | 0.1 |
| 11 | 1.375 | 0.25 | 11 | 1.1 | 0.2 |
| 12 | 1.625 | 0.25 | 12 | 1.3 | 0.2 |
| 13 | 1.875 | 0.25 | 13 | 1.5 | 0.2 |
| 14 | 2.125 | 0.25 | 14 | 1.7 | 0.2 |
| 15 | 2.375 | 0.25 | 15 | 1.9 | 0.2 |
| 16 | 2.75 | 0.5 | 16 | 2.2 | 0.4 |
| 17 | 3.25 | 0.5 | 17 | 2.6 | 0.4 |
| 18 | 3.75 | 0.5 | 18 | 3.0 | 0.4 |
| 19 | 4.25 | 0.5 | 19 | 3.4 | 0.4 |
| 20 | 4.75 | 0.5 | 20 | 3.8 | 0.4 |
| 21 | 5.5 | 1.0 | 21 | 4.4 | 0.8 |
| 22 | 6.5 | 1.0 | 22 | 5.2 | 0.8 |
| 23 | 7.5 | 1.0 | 23 | 6.0 | 0.8 |
| 24 | 8.5 | 1.0 | 24 | 6.8 | 0.8 |
| 25 | 9.5 | 1.0 | 25 | 7.6 | 0.8 |
| 26 | 11.0 | 2.0 | 26 | 8.8 | 1.6 |
| 27 | 13.0 | 2.0 | 27 | 10.4 | 1.6 |
| 28 | 15.0 | 2.0 | 28 | 12.0 | 1.6 |
| 29 | 17.0 | 2.0 | 29 | 13.6 | 1.6 |
| 30 | 19.0 | 2.0 | 30 | 15.2 | 1.6 |
| 31 | 21.5 | 3.0 | 31 | 17.6 | 3.2 |
| 32 | 24.5 | 3.0 | 32 | 20.8 | 3.2 |