# Peer review of "Two months of disdrometer data in the Paris area"

_Earth System Science Data, 2017_

## Referee Comment (RC1) · Anonymous Referee #1 · 19 Jan 2018

The manuscript presents a dataset of drop sizes and velocity for precipitation collected by three disdrometers in January and February 2016 in the Paris area. Additional information about the temperature is also included. The measuring principles of the two types of disdrometers used in this study are described as well as the set-up and the provided datasets. Along with the data, a set of python routines is provided and briefly described for easy data usage. Unfortunately, I wasn't able to test those scripts because it requires python3.

The manuscript fits in the scope of ESSD, but some issues need to be addressed. I recommend taking the following suggestions and comments into account:

1. The authors present a dataset for January and February 2016 obtained in the framework of TARANIS. This is a rather limited dataset of only two month. The authors cite

their article Gires et al. (2017b) in which they use another two month data set from May and June 2016 also obtained in the framework of TARANIS with the same set-up. Therefore, it seems that there is an actual dataset of at least 6 month. Is there a reason, why the authors only provide the data for January and February 2016 in this manuscript and not the whole dataset? According to Gires et al. (2017b) there was a lot of precipitation in May and June 2016 in the Paris area, which makes this dataset even more interesting. I would suggest providing the whole dataset.

2. The links in the html files (Calender_Carnot_1.html, Calender_R_30_sec_Carnot_1.html and Calender_R_5_min_Carnot_1.html) didn't work for me. When clicking on a specific date, an error message appeared stating that the file could not be found. Please check the links!

3. In the introduction the authors describe very briefly for which type of studies the dataset could be useful. I would recommend adding a section at the end of the manuscript that describes the usefulness of the dataset and possibilities for its application in more detail.

Specific comments:

- P.1, L.3: disdrometers measurements –> disdrometer measurements

- P.1, L.8: along with more aggregated one such rain rate –> along with more aggregated ones such as rain rate

- P.1, L.20: such the 2D Video –> such as the 2D Video

- P.2, L.2: equivolimic –> equivolumic

- P.2, L.21: that do not work on the same principle –> operating on different principles

- P.2, L.30: is not the same –> is different

- P.3, L.5: Actually authors found possible –> Actually the authors could

- P.3, L.11: devices –> device

- P.3, L.11: by authors –> by the authors

- P.3, L.18: maide –> made

- P.3, L.25: access the raw data –> access to the raw data

- P.3, L.30: This sentence is confusing. Please rephrase it.

- P.4, L.3: so it user are –> so the users are

- P.4, L.5: What is the resolution of the PWS100 temperature observations?

- P.4, L.8: the the –> the

- P.4, L.20: total depth are –> total depths are

- P.4, L.22: West from disdrometer ... South from disdrometer –> West of the disdrometer... South of the disdrometer

- P. 4, 5 and 6 (description of the data base content): It would be beneficial if the type of data that the folders contain could be added. E.g. P4, L.32: Each folder contains the files of raw data for its disdrometer.

- P.6, L.7: Is it supposed to be Calendars_batiment_Carnot_1 according to the name in the database?

- P.6, L.27: Lhermitte et al., 1988 is missing in the references.

- P.7, L5: semi column –> semicolon

- P.7, L.8: I guess the name of the file is R_5_min_Carnot_1_... in this case?

- P.7, L:11: file –> files

- P.7, L.13: As far as I see, the names of the files in the Daily_data_csv folder in the database only contain the day, not start and end time.

- P.8, L.1: There is two times "1st size class-2nd velocity class" in the enumeration

- P.8, L.4: These files are text file –> These files are text files

- P.9, L.6: each of the day –> each day

- P.9, L.18: The routine is called "extracting_one_event_Carnot_1" in the python script.

I also checked the descriptions in Read_me_v1.txt:

Under point 2) Tools:

- It is v3 and not v2 of the script Tools_data_base_use_v2.py

- The routine is called "extracting_one_event_Carnot_1" in the python script

- A description of the routine exporting_full_matrix_and_T.py is missing

Last sentence: Where can the script Tools_overall_management_"Campaign".py be found? I didn't see it in the python folder.

---

## Referee Comment (RC2) · Anonymous Referee #2 · 3 Mar 2018

**Summary**

This manuscript presents an original data set of raindrop size distribution (DSD), an important piece of information to describe the microstructure of rainfall. The data described in this manuscript come from three optical disdrometers: two Parsivel2 from OTT and a PWS100 from Campbell Scientific. The period of observation cover two winter months in the Paris area in France, with a fair amount of precipitation (between 84 and 104 mm in total depending on the considered disdrometer). In addition to the raw data (size and velocity in a number of classes), the authors give a nice introduction with relevant references, and also provide "derived" quantities like the rain rate for instance, as well as useful tools to browse and visualize the data.

[Figure]

**Recommendation**

The data set is relatively original in the sense that similar data sets (in temperate mid-latitude regions) have been collected and made available to the community, but not with this specific configuration of three collocated instruments among which one of a different type. The fact that these three instruments are collocated, hence making possible the assumption that they sample the same population of raindrops, is attractive to quantify the sampling uncertainties associated to the measurements. The only limitation I see is the rather limited duration of the period of observation: two months is short, and it will be difficult in many related analyses to distinguish peculiar effects due to this short period from more general behaviors. I also have a few minor corrections/suggestions listed below. Overall, I think this is a relevant data set worth to be shared with the community (as is done through the zenodo repository listed in the manuscript). I leave it up to the Editor to decide whether this data set, given its short temporal coverage for rainfall, is worth publication in ESSD or not. I provide in any case a list of relatively minor comments and questions below.

**Specific comments**

1. P.2, l.5: it should be mentioned that assumptions or external information about the scattering properties must be used to derive radar variables from DSD measurements.

2. P.2, l.14: the need for appropriate parameterizations of the DSD in numerical atmospheric models could also be mentioned as an important application of DSD measurements.

3. P.3, l.14: matrixes should be matrices.

4. P.3, l.16: multiplying by 10/9 to compensate for the "gap" in the measurements implies the assumption of homogeneity, this should be mentioned.

5. P.3, l.17: made instead of maide.

6. P.3, Eq.(1): I am not sure about the units to be used in this equation. If $S_{eff}$ is expressed in cm$^2$ as suggested in the paragraph below, then there might be an error in Eq.(1). I think it u $6\pi$ rather than$\pi/6$ (or $600\pi$ if $S_{eff}$ is in mm$^2$). The authors should check...

7. P.3, Eq.(2): $v_i$ is not defined...

8. P.4, l.3: so it user are: rephrase.

9. P.4, l.16: is it 0.1 mm? If so, the authors should add 0 to make it clear.

10. P.4, l.22: the measured amount over the same period of time at the MeteoFrance site would also be relevant to complement the climatological value (that shows that this period is not too specific, at least in terms of rainfall amount over two months...).

11. P.4, end of Section 2.3: in my opinion, some important aspects are not mentioned: in January and February, solid precipitation can occur, was it the case? To this respect, the temperature observations are crucial, but there is a large discrepancy between the two types of instruments (around 3 deg C in the example in in Figure 4, nicely illustrating its importance: according to the Parsivel, it may snow at the end of the day, while it would only be rain according to the PWS100...); The instruments seem to be close to the edge of the roof, raising concerns about turbulence and wakes; what is the direction of the dominant wind at this site? How does it align with the respective orientation of each disdrometer? These are important aspects to clarify to better assess the quality of the data and the possible applications.

12. P.7, l.7: how are treated the possible zeroes when integrating the DSD in time?

---

## Author Comment (AC1) · 29 Mar 2018

First the authors would like to thank the reviewers for their suggestions and careful reading that helped improve the manuscript. Hopefully the changes implemented will satisfy their requirements!

The manuscript presents a dataset of drop sizes and velocity for precipitation collected by three disdrometers in January and February 2016 in the Paris area. Additional information about the temperature is also included. The measuring principles of the two types of disdrometers used in this study are described as well as the set-up and the provided datasets. Along with the data, a set of python routines is provided and briefly described for easy data usage. Unfortunately, I wasn't able to test those scripts because it requires python3.
The manuscript fits in the scope of ESSD, but some issues need to be addressed. I recommend taking the following suggestions and comments into account:
1. The authors present a dataset for January and February 2016 obtained in the frame-work of TARANIS. This is a rather limited dataset of only two month. The authors cite their article Gires et al. (2017b) in which they use another two month data set from May and June 2016 also obtained in the framework of TARANIS with the same set-up. Therefore, it seems that there is an actual dataset of at least 6 month. Is there a reason, why the authors only provide the data for January and February 2016 in this manuscript and not the whole dataset? According to Gires et al. (2017b) there was a lot of precipitation in May and June 2016 in the Paris area, which makes this dataset even more interesting. I would suggest providing the whole dataset.

This issue was also pointed out by the other reviewer. As mentioned in the title, the paper dataset contains two months of data. It corresponds to two months of data with a cumulative depth consistent with the local climatological average. No extreme events were recorded, i.e. the maximums observed at both 5 min and 30 min have return periods smaller than one month. Such "common" events are notably relevant for urban water managers because they correspond to ones for which they should be able to fully decontaminate storm water before release in the natural environment. Furthermore over this range of value the devices are expected to be reliable. Following the reviewers remark, this point was clarified in the presentation of the measurement period. Moreover the devices and additional ones are still operating and collecting data. Hence some additional data will be made available through our website (https://hmco.enpc.fr/portfolio-archive/taranis-observatory/) which already contains links to the calendars for the various past and ongoing (daily updates) measurement campaigns in which the devices were used. Following the reviewers remarks, this was clarified at the end of section 4.

2. The links in the html files (Calender_Carnot_1.html, Calender_R_30_sec_Carnot_1.html and Calender_R_5_min_Carnot_1.html) didn't work for me. When clicking on a specific date, an error message appeared stating that the file could not be found. Please check the links!

Actually, I do not really understand why because I downloaded the file Calendars_batiment_Carnot_1.zip from zenodo, unzipped it and it worked... It might be that your file Calendar_R_30_sec_Carnot_1.html somehow was not located in the same folder as the folders "Quicklooks", "Data_5_min" and "Data_30_sec". Because indeed the links are "relative" and assume this. Following your remark, this was clarified in the manuscript (subsection 3.1). Please let me know if this was that.

3. In the introduction the authors describe very briefly for which type of studies the

dataset could be useful. I would recommend adding a section at the end of the manuscript that describes the usefulness of the dataset and possibilities for its application in more detail.

There are numerous applications of the DSD as mentioned in the introduction. Authors do not believe that it is the purpose of this data paper to go into too much details and that citing relevant papers is sufficient in this context. Nevertheless the reviewer is right that an overall explanation on the fact that all the discussed quantities are basically derived as integrals of the DSD was missing and is now added for clarification.

Specific comments:
- P.1, L.3: disdrometers measurements –> disdrometer measurements
- P.1, L.8: along with more aggregated one such rain rate –> along with more aggregated ones such as rain rate
- P.1, L.20: such the 2D Video –> such as the 2D Video
- P.2, L.2: equivolimic –> equivolumic
- P.2, L.21: that do not work on the same principle –> operating on different principles
- P.2, L.30: is not the same –> is different
- P.3, L.5: Actually authors found possible –> Actually the authors could
- P.3, L.11: devices –> device
- P.3, L.11: by authors –> by the authors
- P.3, L.18: maide –> made
- P.3, L.25: access the raw data –> access to the raw data
- P.3, L.30: This sentence is confusing. Please rephrase it.
- P.4, L.3: so it user are –> so the users are
- P.4, L.5: What is the resolution of the PWS100 temperature observations?
- P.4, L.8: the the –> the
- P.4, L.20: total depth are –> total depths are
- P.4, L.22: West from disdrometer . . . South from disdrometer –> West of the disdrometer. . . South of the disdrometer

This was corrected. Thank you for your careful reading !

- P. 4, 5 and 6 (description of the data base content): It would be beneficial if the type of data that the folders contain could be added. E.g. P4, L.32: Each folder contains the files of raw data for its disdrometer.

Actually, the files are described in the corresponding sub-section. You would like to add some information in the database summary structure ?

- P.6, L.7: Is it supposed to be Calendars_batiment_Carnot_1 according to the name in the database?

Indeed there is a mismatch between the paper and the data base. It was corrected in the data base.

- P.6, L.27: Lhermitte et al., 1988 is missing in the references.
- P.7, L5: semi column –> semicolon
- P.7, L.8: I guess the name of the file is R_5_min_Carnot_1_... in this case?
- P.7, L:11: file –> files

This was corrected.

- P.7, L.13: As far as I see, the names of the files in the Daily_data_csv folder in the database only contain the day, not start and end time.

Indeed, the format was corrected in the text.

- P.8, L.1: There is two times "1st size class-2nd velocity class" in the enumeration
- P.8, L.4: These files are text file –> These files are text files
- P.9, L.6: each of the day –> each day
- P.9, L.18: The routine is called "extracting_one_event_Carnot_1" in the python script.

This was corrected. Again, thank you for your careful reading !

I also checked the descriptions in Read_me_v1.txt:
Under point 2) Tools:
- It is v3 and not v2 of the script Tools_data_base_use_v2.py
- The routine is called "extracting_one_event_Carnot_1" in the python script
- A description of the routine exporting_full_matrix_and_T.py is missing

This was corrected.

Last sentence: Where can the script Tools_overall_management_"Campaign".py be found? I didn't see it in the python folder.

Indeed it was a sentence remaining from the files I used to actually collected the data. The function "Generation_daily_data_python_Carnot_1" was added to the file "Tools_data_base_use_v3.py";
This was corrected in the read_me file. As a consequence at the final stage of the review, I will also update the database with a new doi.

---

## Author Comment (AC2) · 29 Mar 2018

First the authors would like to thank the reviewers for their suggestions and careful reading that helped improve the manuscript. Hopefully the changes implemented will satisfy their requirements!

This manuscript presents an original data set of raindrop size distribution (DSD), an important piece of information to describe the microstructure of rainfall. The data described in this manuscript come from three optical disdrometers: two Parsivel2 from OTT and a PWS100 from Campbell Scientific. The period of observation cover two winter months in the Paris area in France, with a fair amount of precipitation (between 84 and 104 mm in total depending on the considered disdrometer). In addition to the raw data (size and velocity in a number of classes), the authors give a nice introduction with relevant references, and also provide "derived" quantities like the rain rate for instance, as well as useful tools to browse and visualize the data.

Recommendation
The data set is relatively original in the sense that similar data sets (in temperate midlatitude regions) have been collected and made available to the community, but not with this specific configuration of three collocated instruments among which one of a different type. The fact that these three instruments are collocated, hence making possible the assumption that they sample the same population of raindrops, is attractive to quantify the sampling uncertainties associated to the measurements. The only limitation I see is the rather limited duration of the period of observation: two months is short, and it will be difficult in many related analyses to distinguish peculiar effects due to this short period from more general behaviors. I also have a few minor corrections/suggestions listed below. Overall, I think this is a relevant data set worth to be shared with the community (as is done through the zenodo repository listed in the manuscript). I leave it up to the Editor to decide whether this data set, given its short temporal coverage for rainfall, is worth publication in ESSD or not. I provide in any case a list of relatively minor comments and questions below.

This issue was also pointed out by the other reviewer. As mentioned in the title, the paper dataset contains two months of data. It corresponds to two months of data with a cumulative depth consistent with the local climatological average. No extreme events were recorded, i.e. the maximums observed at both 5 min and 30 min have return periods smaller than one month. Such "common" events are notably relevant for urban water managers because they correspond to ones for which they should be able to fully decontaminate storm water before release in the natural environment. Furthermore over this range of value the devices are expected to be reliable. Following the reviewers remark, this point was clarified in the presentation of the measurement period. Moreover the devices and additional ones are still operating and collecting data. Hence some additional data will be made available through our website (https://hmco.enpc.fr/portfolio-archive/taranis-observatory/) which already contains links to the calendars for the various past and ongoing (daily updates) measurement campaigns in which the devices were used. Following the reviewers remarks, this was clarified at the end of section 4.

Specific comments
1. P.2, l.5: it should be mentioned that assumptions or external information about the scattering properties must be used to derive radar variables from DSD measurements.

Indeed, you are right and this was clarified.

2. P.2, l.14: the need for appropriate parameterizations of the DSD in numerical atmospheric models could also be mentioned as an important application of DSD measurements.

This is indeed another possible application, which is now mentioned in the introduction.

3. P.3, l.14: matrixes should be matrices.

This was corrected.

4. P.3, l.16: multiplying by 10/9 to compensate for the "gap" in the measurements implies the assumption of homogeneity, this should be mentioned.

You are indeed right, and this was clarified in the text.

5. P.3, l.17: made instead of maide.

This was corrected.

6. P.3, Eq.(1): I am not sure about the units to be used in this equation. If $S_{eff}$ is expressed in cm $^2$ as suggested in the paragraph below, then there might be an error in Eq.(1). I think it u $6\pi$ rather than $\pi/6$ (or $600\pi$ if $S_{eff}$ is in mm $^2$). The authors should check...

This was checked and units clarified.

7. P.3, Eq.(2): $v_i$ is not defined...

It is actually defined at the beginning of section 2.2

8. P.4, l.3: so it user are: rephrase.

This was corrected, thank you for your careful reading.

9. P.4, l.16: is it 0.1 mm? If so, the authors should add 0 to make it clear.

Yes, and the "0" was added.

10. P.4, l.22: the measured amount over the same period of time at the MeteoFrance site would also be relevant to complement the climatological value (that shows that this period is not too specific, at least in terms of rainfall amount over two months...).

Such information is not available to the authors, so it was not added.

11. P.4, end of Section 2.3: in my opinion, some important aspects are not mentioned: in January and February, solid precipitation can occur, was it the case? To this respect, the temperature observations are crucial, but there is a large

discrepancy between the two types of instruments (around 3 deg C in the ex-
ample in in Figure 4, nicely illustrating its importance: according to the Parsivel,
it may snow at the end of the day, while it would only be rain according to the
PWS100...); The instruments seem to be close to the edge of the roof, raising
concerns about turbulence and wakes; what is the direction of the dominant wind
at this site? How does it align with the respective orientation of each disdrom-
eter? These are important aspects to clarify to better assess the quality of the
data and the possible applications.

The reviewer is right, and discussion was expanded. Indeed the devices are on edge of the roof, and it might cause some issues notably with regards to the wind. Hence the installation of two perpendicular devices. This was clarified in the text. With regards to solid precipitation, basically no snow was recorded during this period except a little on February 18. Snow can be identified through the recorded fall velocity of the hydrometeors. indeed for a similar size, snowflakes will fall much slower than rain drops

12. P.7, l.7: how are treated the possible zeroes when integrating the DSD in time?

Authors do not understand the point, since p.7 l.7 is not discussing DSD...